# Energy Balance, Hormonal Status, and Military Performance in Strenuous Winter Training

**DOI:** 10.3390/ijerph20054086

**Published:** 2023-02-24

**Authors:** Tarja Nykänen, Tommi Ojanen, Jani P. Vaara, Kai Pihlainen, Risto Heikkinen, Heikki Kyröläinen, Mikael Fogelholm

**Affiliations:** 1Army Academy, Finnish Defence Forces, 53600 Lappeenranta, Finland; 2Finnish Defence Research Agency, Finnish Defence Forces, 04310 Tuusula, Finland; 3Department of Leadership and Military Pedagogy, National Defence University, Finnish Defence Forces, 00861 Helsinki, Finland; 4Defence Command, Finnish Defence Forces, 00130 Helsinki, Finland; 5Statistical Analysis Services, Analyysitoimisto Statisti Oy, 40720 Jyväskylä, Finland; 6Faculty of Sport and Health Sciences, University of Jyväskylä, 40114 Jyväskylä, Finland; 7Department of Food and Nutrition, University of Helsinki, 00014 Helsinki, Finland

**Keywords:** energy deficit, physical performance, survival training, soldier

## Abstract

Severe energy deficit may impair hormonal regulation and physical performance in military trainings. The aim of this study was to examine the associations between energy intake, expenditure, and balance, hormones and military performance during a winter survival training. Two groups were studied: the FEX group (*n* = 46) had 8-day garrison and field training, whereas the RECO group (*n* = 26) had a 36-h recovery period after the 6-day garrison and field training phase. Energy intake was assessed by food diaries, expenditure via heart rate variability, body composition by bioimpedance, and hormones by blood samples. Strength, endurance and shooting tests were done for evaluating military performance. PRE 0 d, MID 6 d, POST 8 d measurements were carried out. Energy balance was negative in PRE and MID (FEX −1070 ± 866, −4323 ± 1515; RECO −1427 ± 1200, −4635 ± 1742 kcal·d^−1^). In POST, energy balance differed between the groups (FEX −4222 ± 1815; RECO −608 ± 1107 kcal·d^−1^ (*p* < 0.001)), as well as leptin, testosterone/cortisol ratio, and endurance performance (*p* = 0.003, *p* < 0.001, *p* = 0.003, respectively). Changes in energy intake and expenditure were partially associated with changes in leptin and the testosterone/cortisol ratio, but not with physical performance variables. The 36-h recovery restored energy balance and hormonal status after strenuous military training, but these outcomes were not associated with strength or shooting performance.

## 1. Introduction

Winter survival training is one of the most demanding forms of field training for soldiers, in which they are exposed to various internal and external stressors. Energy deficit and sleep deprivation, combined with mentally demanding tasks in harsh environments, impair hormonal regulation and physical performance [1,2,3,4,5]. Energy expenditure is elevated due to prolonged low- to high-intensity military tasks, personal load carriage, limited sleep, and extreme weather conditions in the field [6,7,8]. Energy intake is often decreased because of insufficient energy content of field rations, intentional restriction of food intake, menu fatigue, or bad mood [6,9]. Furthermore, logistical problems or an extra load of food rations can diminish food supply in field trainings [7]. Thus, imbalance in energy intake and expenditure cause negative energy balance, which can occur for several days or weeks in military field trainings [6].

Severe energy deficit impairs hormonal regulation and disturbs homeostasis in harsh environments [10,11]. Appetite-mediating hormones, leptin and ghrelin, react in acute caloric deprivation [12] and in response to loss of body mass [13]. Catabolic hormones, such as cortisol, are increased and anabolic hormones (e.g., testosterone) are decreased in a negative energy balance, but hormonal changes can be recovered during training, when more energy is available [14]. Hamarsland et al. [2] observed that serum testosterone and cortisol concentrations stayed imbalanced after 24- and 72-h recovery, and cortisol levels were still elevated after 1 week of arduous military course. Nevertheless, studies have not clearly demonstrated, if a short and active recovery period could normalize hormonal status enough to maintain military performance in the field.

Declines in lower-body power and strength after arduous military training were associated with energy deficit and duration of training, while greater total negative energy balance was related to the impairment of lower-body performance [3]. Fewer studies have examined endurance capacity, but impairment of aerobic, but not anaerobic, performance has been observed [7]. The 7-d arduous military course, with energy and sleep deficit, resulted in a decrease in maximal strength [2]. After a 2-d recovery, no significant changes were observed in strength tests, but after 7 d of recovery, maximal strength values, except for counter movement jump, recovered close to pre-values, indicating hormonal and physical recovery [2,15]. In addition, energy deficit has been shown to alter shooting performance during training, especially in a standing position [16]. It is still unclear, whether energy variables are associated with a decline in physical and military performance in strenuous field training.

Thus, the purpose of the present study was (a) to observe energy intake, energy expenditure, and energy balance in strenuous field exercise; (b) to study associations of these energy variables with hormonal status (leptin, ghrelin, testosterone/cortisol ratio (T/C ratio)), and strength, endurance and shooting performance; and (c) to investigate effects of a 36-h recovery on these outcomes in soldiers.

## 2. Materials and Methods

Sixty-eight male soldiers volunteered for the study, and they were divided into two groups (FEX = field exercise, RECO = recovery) according to their platoon. The FEX group (*n* = 42) had garrison and field military training, whereas the RECO group (*n* = 26) had a 36-h recovery period after the military field training phase. Randomization was not possible, given that survival training was a part of their military service and implemented by platoons. For the field exercise group, more participants were selected, because a higher attrition rate was expected from this group. The present study was part of a larger multidisciplinary research project, authorized by the Finnish Defence Forces (AO1720). The ethical review of the study was granted by the Scientific and Ethical Committee of the Helsinki University Hospital Research (HUS/900/2018). All participants were informed of the experimental design, methods, benefits, and possible risks prior to signing an informed consent document. Characteristics of participants are shown in Table 1.

The study protocol is presented in Table 2. The training was divided into three phases: (1) 3-day preparation period at the garrison; (2) 2-day survival training in the field; (3) 2-day survival training (FEX) or recovery (RECO) period.

The study was conducted in Lapland, north from the Arctic Circle (location 67°24′54” N, 26°35′26” E) in early April. Measurements were carried out at three time-points: at baseline (PRE), after phase 2 (MID), and after phase 3 (POST). During phase 1, participants were educated on the theory and practice of survival training. The workload was moderate, 3–4 meals were served at the canteen, and sleeping was arranged similar to normal service. During phase 2, all participants started their field training period where the workload was high, while energy intake and sleep were restricted. Additionally, military tasks were stressful and arduous, partially due to snow (depth 80–100 cm). Changes in temperature were typical for late winter: mean daily temperature varied from −0.3 °C to −4.7 °C; min −10.5 °C at night; max 5.4 °C in the daytime. Since participants had started their conscript service earlier, they were adapted to local weather conditions. Moreover, the temperatures were not extreme considering any location in Finland during wintertime. Personal equipment was carried by backpacks and pulks (extra load 23–32 kg). During phase 3, the FEX group continued their field training as described, but the RECO group was moved to a temporary accommodation, where facilities were similar to the garrison (3–4 meals served, extra snacks, good sleeping facilities, bathroom, and sauna). In addition, low-intensity activities (ball games and stretching) were supervised to help recovery.

Body composition and blood samples were carried out in a fasting state, early in the morning, with the same protocol for each measurement point. A segmental multi-frequency bioimpedance analysis (Inbody 720/770, Biospace, Soul, Republic of Korea) was used to evaluate body mass and fat-free mass. Participants wore underwear during the assessment, and they were advised to urinate before the measurement.

For leptin, ghrelin, testosterone, and cortisol analysis, blood samples were collected into VenoSafe plastic tubes (VenoSafe^®^, Terumo Europe, Leuven, Belgium) containing silica gel. Samples were centrifuged (3500 rpm, 10 min), then the supernatant (serum) was collected and frozen for later analysis. Serum leptin and ghrelin levels were determined with an ELISA-kit immunoassay system (Dynex DS 2, Dynex Technologies, Chantilly, VA, USA). Testosterone and cortisol levels were acquired via Immulite immunoassay analyzer (Siemens Immulite 2000 XPI, Siemens Healthcare, Malvern, PA, USA). The sensitivity and inter-assay coefficient of variation for these assays were: 0.2 ng/mL and 6.1% for leptin; 0.6 pg/mL and 17.3% for ghrelin; 0.5 nmol/L and 7.8% for testosterone; and 5.5 nmol/L and 6.5% for cortisol.

Total energy expenditure and exercise energy expenditure were estimated via continuous heart rate variability recording (Firstbeat Bodyguard 2, Firstbeat Technologies Oy, Jyväskylä, Finland). Participants wore a two-electrode portable device during the study period. For energy availability, days 5 and 7 were chosen to get entire 24-h data and to synchronize energy expenditure with energy intake values. Exercise energy expenditure was objectively determined from heart rate variability data by the manufacturer’s analysis. Continuous heart rate monitoring has been reported to be sufficiently accurate in light- and moderate-intensity activities [17]. A strong correlation (r = 0.85–0.98; *p* < 0.05) between gold-standard energy expenditure measurement and energy expenditure evaluation has been observed in graded tests [18].

Energy intake was estimated with pre-filled food diaries by a software program (Fineli, National Food Composition Database, Finland). Meals served at phase 1 and 3, as well as food items at phases 2 and 3, were known beforehand, so diaries were pre-filled and participants marked the time and the amount of food consumed. During phase 2, restricted field ratios consisted of one protein bar (226 kcal), eight crackers (306 kcal), and two lunch meal rations (mean 661 kcal/portion). Representative days (5 and 7) were analyzed further in relation to the other energy variables.

Maximal isometric force of the upper- and lower-body extremities was measured bilaterally by a leg and bench press dynamometer (Faculty of Sport and Health Sciences, University of Jyväskylä, Finland). In bench and leg press, positions were adjusted as previously described [19,20]. Participants were instructed to produce maximal force as fast as possible with verbal encouragement by the testing personnel. One trial attempt was allowed before the two test trials, with minimum 60 s recovery between the trials. The best performance was selected for further analysis.

A standing long jump was used to measure explosive force production of the lower extremities [21] on a specifically designed gym mat (Fysioline Co., Tampere, Finland). Before testing, the participants were instructed on the correct technique, and they performed 2–3 warm-up jumps. The participants were advised to jump (horizontally) forward as far as possible from a standing position and land bilaterally without falling backward. The best of the three jumps was utilized, measuring from the start line to the landing point.

A seated medicine ball throw was measured for assessing explosive force production of the upper body [22]. The participants sat in an upright position on the floor with their legs fully extended and back kept against the vertical wall throughout the test. The 3-kg medicine ball was held with both hands in front of their chest, with their forearms positioned parallel to the ground. The participant threw the ball vigorously as far forward as possible, while maintaining their back against the wall. The distance from the wall to landing point of the medicine ball was recorded. The best result out of the three trials was used in the analysis. The participants were allowed to have at least three training throws before the test measurements.

Sit-ups and pull-ups were conducted to assess muscular endurance by counting maximal repetitions in one minute. Sit-ups were used to assess abdominal and hip flexor performance [23], and push-ups were used to measure performance of the arm and shoulder extensor muscles [24]. Technical advice was given before the tests and incorrect repetitions were excluded. All strength tests were transformed to z-scores and the mean of values formed “strength index”.

Endurance performance was measured by a 20-m shuttle run test [25]. Participants were advised to run 20 m back and forth with accelerating pace as long as possible. The test was finished when participants were not able to keep the given pace or voluntarily dropped out. Maximal oxygen uptake values were calculated from the test results, as previously described [26].

Shooting accuracy was estimated by an optical infra-red weapon system (Eko-Aims Oy, Ylämylly, Finland). Ten shots were given in prone and standing positions to the target 10 m away from the shooting line, in an indoor hall. The sum of ten shots from both positions was recorded for analysis. The sum variable “shooting index” was aggregated from these results.

Physical and shooting performance, as well as blood biomarkers, were set as dependent variables, time × group interaction and energy variables as independent variables, and body mass as the confounding factor. Time × group interactions were tested with F-tests based on the Satterthwaite method, using the lmerTest R-package. A linear mixed effect model was used to estimate changes between groups over the studied period. Sample size varied depending on the variables tested. In addition, the specific sample sizes are being reported in the tables and figures. Pairwise comparisons were performed using a Tukey’s test. Logarithmic transformations were done when the distribution was positively skewed (leptin, ghrelin, testosterone-cortisol ratio). Non-parametric Mann–Whitney U-tests were used to verify the linear mixed effect model when residuals were not normally distributed (strength index, testosterone-cortisol ratio). Pairwise Pearson correlations were calculated to estimate associations between energy variables and strength, and the shooting index, endurance performance and hormonal status. All statistical analyses were performed using R v. 3.6.3 (2020, R Foundation for Statistical Computing, Vienna, Austria). Values are presented as the means ± standard deviations and statistical significance was set at *p* < 0.05.

## 3. Results

The mean body mass reduced in the FEX group from PRE to MID (*p* < 0.001), and was then maintained at POST (74.4 ± 10.7, 72.9 ± 9.8, 72.6 ± 9.6 kg, respectively), but in RECO, body mass first decreased in MID (*p* < 0.001), but then recovered towards the baseline (*p* < 0.001) (78.2 ± 9.7, 74.5 ± 9.2, 77.1 ± 8.6 kg, respectively). No differences were found between the groups. Skeletal muscle mass dropped in MID and increased in POST in both groups (FEX 36.2 ± 4.6, 35.9 ± 4.8, 36.9 ± 4.8 kg; RECO 38.0 ± 3.7, 37.1 ± 3.1, 38.6 ± 3.2 kg; *p* < 0.001 PRE–MID, MID–POST both). Fat mass decreased in the FEX group at all measurement points (10.6 ± 5.0, 9.4 ± 3.5, 7.6 ± 3.0 kg; *p* < 0.001 PRE–MID, MID–POST). In the RECO group, fat mass first reduced and then stabilized during the recovery period (11.4 ± 5.4, 9.0 ± 5.5, 8.7 ± 4.7 kg; *p* < 0.001 PRE–MID, MID–POST). More results have been presented in the previous study [10].

Energy intake was low, especially in the MID measurement. The RECO group received more energy than the FEX group in the POST measurement. At that time, the RECO group was transferred into an accommodation, where energy expenditure lowered and meals were served similar to the garrison. Energy expenditure values exceeded the energy intake values throughout the study, even in garrison-like circumstances. These two outcomes caused a negative energy balance in all measurement points. All results of the energy variables are shown in Figure 1. Hormonal responses are presented in Figure 2. Differences between the groups were found in leptin at MID and POST, and also in the testosterone/cortisol ratio at POST.

Absolute results of physical fitness and shooting tests are shown in Table 3. Significant differences between the groups were found in the 20-m shuttle run test, where the RECO group improved its performance in POST, and in prone shooting, in which the groups differed at all measurement points. Push-ups and shooting in standing position differed at baseline, but the differences disappeared in MID and POST between the groups.

Sum variable “strength index” decreased from PRE to MID in both groups (FEX *p* < 0.001, RECO *p* = 0.014), and increased from MID to POST only in RECO (*p* < 0.001). Strength index differed between the groups (*p* = 0.025) at baseline, but the difference disappeared in later measurements. Endurance performance (20-m shuttle run) impaired at MID in both (FEX *p* < 0.001, RECO *p* = 0.007), but the RECO group improved its performance at POST compared to MID (*p* = 0.008). The shooting index decreased only in the FEX group from MID to POST (*p* = 0.0003), but it differed between the groups at all measurement points (*p* < 0.001, *p* = 0.003, and *p* < 0.001 in PRE, MID, POST, respectively), as seen in prone shooting results alone.

The time × group interactions were found for leptin (*p* < 0.001) and strength index (*p* < 0.001), but not between any other variables. Furthermore, energy expenditure (*p* = 0.038) and energy intake (*p* = 0.028) were significant predictors for leptin.

A strong, positive association was found between changes in energy balance and changes in the shooting index (r = 0.786, *p* = 0.002) in the FEX group from MID to POST, whereas no other significant associations were found between energy variables and the strength index, endurance performance, or shooting index. Statistically significant associations between changes in energy intake and energy expenditure, and changes in hormones are presented in Table 4. The change in energy balance was not associated with any of these hormonal variables.

## 4. Discussion

Energy balance was negative during survival training as a result of low energy intake and high energy expenditure. Changes in energy variables were associated with changes in leptin and testosterone/cortisol ratio, but not in physical performance. Energy and hormonal status were partially normalized during the 36-h recovery period.

A negative energy balance is a typical characteristic of survival training [6,27]. The magnitude of energy deficit depends on the duration of training, the amount of daily energy deficit, food intake, and total physical activity. In the PRE-measurement, participants lived in a garrison and prepared for the field training phase. Despite the conditions, energy balance was negative in PRE-measurement, probably as a result of biases in energy intake and expenditure estimation. Energy intake was analyzed via self-reported food diaries, so underestimation could at least partially explain the negative energy balance. The preparation included low-intensity tasks with long working hours, which shortened the sleeping time. It is notable that sleep deprivation increases energy expenditure, because wakefulness maintains energy metabolism activated, even in a habitual day [28]. This phenomenon was enhanced during the field training phase, where participants were not able to sleep for more than 1–3 h per day due to military tasks [20].

Energy intake differed at the POST-measurement, because the RECO group received normal meals in their accommodation. This led to differences in energy intake, expenditure and balance at the last measurement point. Although the RECO group should have had rest in their recovery period, their estimated energy expenditure stayed elevated (over 3500 kcal/day). They were instructed to do light activities to enhance their recovery, and sleeping times were normal. Energy expenditure values have been analyzed by heart rate variability measurements. This method has been reported to be valid in treadmill tests, compared to the golden standard method [18]. Hinde et al. [29] agreed that the HRV monitor is also accurate for military purposes, but some biases exist. When energy expenditure estimation is based on the heart rate, changes in the heart rate may interfere with the results. Resting and submaximal heart rate values may be elevated due to short-term overreaching symptoms after strenuous exercise [30]. In this case, survival training, with limited sleep and increased arousal time, may have elevated energy expenditure levels. Despite methodological inaccuracies, a negative energy balance during survival training was confirmed by a decrease in body mass from PRE to MID in both groups. An increase in body mass after the recovery phase in RECO was obvious, when more energy was consumed.

Strength and endurance performance were impaired after the survival training phase, but no associations were found in changes of energy intake, expenditure, or balance with changes in physical performance. However, the 20-m shuttle run test, which assessed endurance performance, differed between the groups in POST, as expected. Although associations with energy variables were not found, overall fatigue and lack of maximal effort may partially explain the difference between the groups and the decrease in aerobic capacity in the FEX group. Participants in the FEX group were exhausted after the survival training phase, and physical fitness tests in an indoor hall gave them the option to relax or even fall asleep between the tests.

An energy deficit exists in military conditions, but the associations with physical or military performance are unclear. We did not find any associations, which can be partially explained by the duration of fitness tests. Most of the tests were completed in a few seconds, which was mainly covered by the ATP and CP storages, and less energy was required from glycogen storages. That could also explain the difference in the 20-m shuttle run test, where the RECO group, in a fed state, could maintain its performance. However, the relation was not found with any of the energy variables. According to a meta-regression of Murphy et al. [3]., muscle performance was not related to a daily energy balance, but was proportional to the total energy balance, taking into account training duration and daily energy balance. O’Leary et al. [7] presented that decreases in muscle strength and power performance were observed in most military studies, but not all. Based on these results and previous literature, the role of energy deficit in physical performance cannot be identified yet. In military conditions, duration of trainings and the magnitude of energy deficit vary, as well as physical fitness test patterns. Furthermore, military tasks often cause neuromuscular fatigue, sleep deficit, and mental stress, which impair physical performance and energy-related factors simultaneously. The latest studies have measured energy availability in a military context, to clarify the energy needs of soldiers, and all of them indicate that soldiers do not reach the optimal level of energy availability [31,32]. For future research, estimating energy availability, together with energy balance, could provide important insights at the individual level, by advancing our understanding of energy metabolism in soldiers.

The shooting index was positively associated with energy balance in the FEX group in the latter part of the study. Group differences occurred in the prone shooting at all measurements, and also in the standing shooting at PRE. Shooting performance is a complex skill, where anthropometric, technical-coordinative, physiological, and psychological factors contribute to the shooting accuracy [33], not only energy-related factors. Overall and neuromuscular fatigue will affect the readiness and alertness of participants. Furthermore, shooting with an infra-red weapon may diminish the focus during shooting compared to real weapons in an outdoor shooting range. Previous literature indicates that shooting performance could be maintained in survival trainings, despite the overall stress of the training [34]. Additionally, sleep deficit can be eliminated with caffeine to improve sighting and triggering in shooting performance [35]. Ojanen et al. [16] found that prone shooting did not change in a 3-w military field training, but positive associations were observed between changes in strength of the lower- and upper-body with changes in standing shooting.

In our study, energy intake and expenditure were related to changes in leptin, ghrelin, and the T/C ratio. Leptin and ghrelin are appetite-mediating hormones, which react in fasted and fed states. Leptin concentration decreases in starvation, whereas hunger and appetite stimulate ghrelin levels [12]. The T/C ratio describes the relation of anabolic and catabolic hormone levels; thus, the ratio increases when the anabolic state is activated and vice versa [36]. Most of the results agree with existing literature [12,37], except for the inverse association between the change in energy intake and the change in T/C ratio. Physiologically, it is well known that if more energy is consumed, more anabolism is observed in the body. Our controversial finding demonstrates that the T/C ratio decreases when more energy has been given. This finding was from the FEX group during the last phase of the study, and in this situation, the overall stress may affect the participants. Therefore, despite the small increase in energy intake, the T/C ratio was impaired. Sleep deprivation can disturb the interpretation of these results, since restricted sleep (4 h) alone resulted in a decrease in leptin and an increase in ghrelin, without energy deficit [37]. Thus, more research is needed to differentiate stress factors and their contribution to military performance in harsh environments.

A 36-h recovery seemed to increase leptin and the T/C ratio at POST, indicating hormonal recovery. Values did not reach baseline levels, which agrees with previous studies, where total recovery was observed in 7 days [2,15]. Short-time recovery improved endurance performance and the strength index in the RECO group. Such a short and active recovery may be a potential model for maintaining military performance in long-term military exercises.

Some challenges exist in measuring the energy intake, energy expenditure, and body composition: an accurate method for energy expenditure estimation is expensive or time-consuming (e.g., doubly labeled water technique), and energy intake measurement by food diaries often include biases (i.e., underestimation, inaccuracies, and forgetfulness) [38]. Thus, inaccuracies in energy intake and expenditure estimation produce errors in energy balance values. Body composition assessment by bioimpedance analysis may overestimate fat-free mass [39]. Furthermore, most of these measurements should be carried out in a laboratory to get the most accurate data, but in military conditions, this may not be possible. For military purposes, it is valuable to get data from the field and from the unique study protocol.

## 5. Conclusions

An 8-d survival training caused a remarkable energy deficit and response in hormonal status, but energy intake, expenditure, or balance were not related to the physical performance of soldiers. A 36-h recovery restored the energy balance and hormonal status after strenuous military training, but these outcomes were not associated with strength or shooting performance. A short recovery can return the energy and hormonal balance to near-normal levels, which can yield a better performance of military tasks. Most recent studies in this area have examined the short-term effects of strenuous military training, but the long-term physiological responses of intermittent energy deficit and recovery in military deployment require further research [7].

## Figures and Tables

**Figure 1 ijerph-20-04086-f001:**
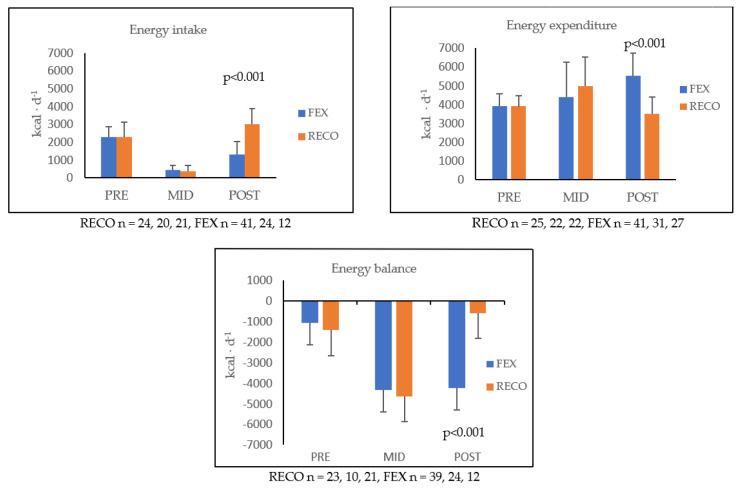
Mean (± SD) energy intake (EI), energy expenditure (EE), and energy balance (EB) during the training in the RECO and FEX groups. RECO = recovery, FEX = field exercise. Statistical significances are presented between the groups.

**Figure 2 ijerph-20-04086-f002:**
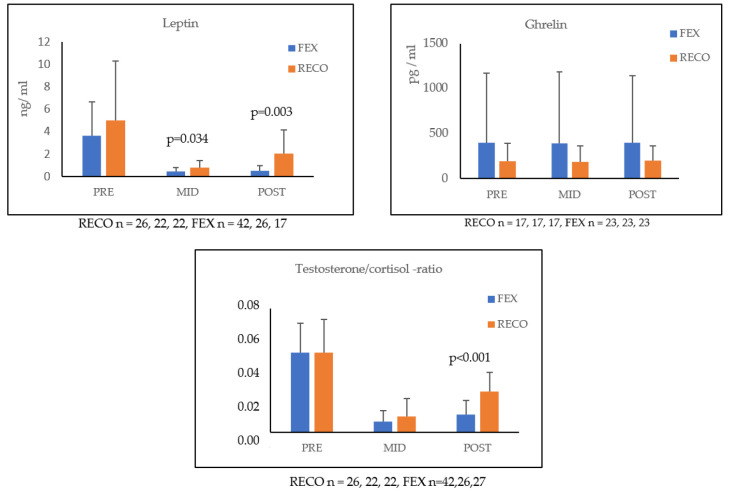
Mean (±SD) leptin, ghrelin, and testosterone/cortisol ratio concentrations during the training in the recovery (RECO) and field exercise (FEX) groups. Statistical significances are presented between the groups.

**Table 1 ijerph-20-04086-t001:** Characteristics of participants in the RECO and FEX groups at baseline. Values are the means ± SD. No statistically significant differences between the groups were found.

Group	*n*	Age (Years)	Height (cm)	Body Mass (kg)	BMI (kg/m^2^)
RECO	26	19.7 ± 1.2	181 ± 6	78.2 ± 9.6	23.9 ± 2.7
FEX	42	19.6 ± 0.8	179 ± 7	74.4 ± 10.7	23.1 ± 2.8

**Table 2 ijerph-20-04086-t002:** Study protocol and the used method for each variable below. x refers to measurement day.

Days	PRE Day 1	Day 2	Day 3	Day 4	Day 5	MID Day 6	Day 7	POST Day 8
Task	Education in garrison			Field exercise		Recovery in RECO Field exercise in FEX		
Energy expenditure	x				x		x	
Heart rate variability								
Energy intake	x				x		x	
Pre-filled food diaries								
Blood samples	x					x		x
Immunoassay								
Body composition	x					x		x
Bioimpedance								
Fitness tests	x					x		x
Strength and endurance								
Shooting test	x					x		x
Prone and standing								

**Table 3 ijerph-20-04086-t003:** Mean (±SD) physical fitness tests and shooting performance results during the training in the field exercise (FEX) and recovery (RECO) groups. Statistical significances are presented between the groups. Number of participants are shown in brackets. VO_2_max refers to maximal oxygen uptake.

Physical and Shooting Performance	Group	PRE	MID	POST
Maximal isometric force, lower (kg)	FEX	88 ± 10 (42)	82 ± 11 (28)	85 ± 12 (26)
	RECO	83 ± 11 (25)	79 ± 11 (22)	84 ± 11 (21)
Maximal isometric force, upper (kg)	FEX	312 ± 89 (42)	312 ± 88 (28)	336 ± 103 (26)
	RECO	314 ± 74 (25)	296 ± 79 (22)	321 ± 66 (21)
Standing long jump (cm)	FEX	225 ± 22(41)	222 ± 22 (27)	221 ± 18 (24)
	RECO	217 ± 16 (24)	217 ± 16 (22)	218 ± 15 (21)
Medicine ball throw (cm)	FEX	611 ± 74 (42)	590 ± 68 (28)	575 ± 68 (26)
	RECO	595 ± 58 (24)	585 ± 51 (22)	609 ± 47 (21)
Push-ups (reps/min)	FEX	39 ± 13 (41)	30 ± 14 (27)	34 ± 12 (24)
	RECO	32 ± 11 (24) *p* = 0.029	27 ± 13 (22)	36 ± 10 (21)
Sit-ups (reps/min)	FEX	43 ± 9 (41)	40 ± 10 (27)	39 ± 9 (24)
	RECO	39 ± 6 (24)	39 ± 8 (22)	41 ± 7 (21)
20-m shuttle run for VO_2_max (ml·kg^−1^·min^−1^)	FEX	46 ± 5 (24)	39 ± 7 (27)	36 ± 12 (23)
	RECO	45 ± 5 (41)	41 ± 8 (21)	45 ± 5 (20) *p* = 0.003
Shooting, prone (points)	FEX RECO	86 ± 10 (24) 93 ± 6 (26) *p* < 0.001	88 ± 7 (28) 93 ± 3 (22) *p* = 0.001	76 ± 9 (26) 91 ± 6 (21) *p* < 0.001
Shooting, standing (points)	FEX	62 ± 10 (42)	63 ± 11 (28)	64 ± 12 (26)
	RECO	69 ± 11 (26) *p* = 0.016	68 ± 11 (22)	69 ± 11 (21)

**Table 4 ijerph-20-04086-t004:** Associations of changes in energy intake, energy expenditure, and energy with changes in leptin, ghrelin, and testosterone/cortisol ratio (T/C ratio). Statistically significant associations have been bolded.

		Leptin	Ghrelin	T/C Ratio
PRE-POST changes				
Energy intake	FEX	r = −0.089, *p* = 0.784	r = −0.003, *p* = 0.413	r = −0.214, *p* = 0.504
	RECO	r = 0.099, *p* = 0.687	r = −0.215, *p* = 0.46	r = −0.035, *p* = 0.888
Energy expenditure	FEX	**r = −0.566, *p* = 0.003**	r = −0.131, *p* = 0.552	r = −0.231, *p* = 0.257
	RECO	r = −0.007, *p* = 0.977	r = 0.028, *p* = 0.915	r = −0.145, *p* = 0.145
Energy balance	FEX	r = 0.398, *p* = 0.2	r = 0.104, *p* = 0.774	r = 0.104, *p* = 0.747
	RECO	r = 0.197, *p* = 0.419	r = −0.466, *p* = 0.093	r = −0.387, *p* = 0.102
PRE-MID changes				
Energy intake	FEX	r = −0.302, *p* = 0.173	r = −0.35, *p* = 0.12	r = 0.014, *p* = 0.951
	RECO	r = −0.384, *p* = 0.116	r = −0.123, *p* = 0.689	**r = 0.479, *p* = 0.044**
Energy expenditure	FEX	r = −0.059, *p* = 0.775	r = 0.388, *p* = 0.067	r = −0.366, *p* = 0.066
	RECO	r = −0.27, *p* = 0.225	r = −0.001, *p* = 0.996	r = −0.017, *p* = 0.94
Energy balance	FEX	r = −0.187, *p* = 0.405	r = −0.337, *p* = 0.135	r = 0.245, *p* = 0.328
	RECO	r = −0.181, *p* = 0.473	r = −0.163, *p* = 0.595	r = 0.15, *p* = 0.506
MID-POST changes				
Energy intake	FEX	r = −0.086, *p* = 0.814	r = 0.121, *p* = 0.74	**r = −0.668, *p* = 0.035**
	RECO	**r = −0.560**, ***p* = 0.013**	r = 0.317, *p* = 0.269	r = −0.284, *p* = 0.254
Energy expenditure	FEX	**r = −0.512, *p* = 0.011**	r = 0.214, *p* = 0.326	r = −0.375, *p* = 0.071
	RECO	r = −0.165, *p* = 0.269	r = −0.238, *p* = 0.358	r = −0.236, *p* = 0.303
Energy balance	FEX	r = 0.43, *p* = 0.215 r = −0.067, *p* = 0.784	r = −0.275, *p* = 0.441 r = 0.489, *p* = 0.076	r = −0.25, *p* = 0.485 r = −0.027, *p* = 0.916
	RECO			

## Data Availability

The data presented in this study are available on request from the corresponding author. The data are not publicly available due to privacy.

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
