# Peer review of "Energy Balance, Hormonal Status, and Military Performance in Strenuous Winter Training"

_ijerph, 2023, doi:10.3390/ijerph20054086_

Round 1

Reviewer 1 Report

The research is undoubtedly interesting, relevant and corresponds to the direction of the journal. At the same time, the reviewer has a number of comments and suggestions.

The text repeatedly states that 46 soldiers were examined (L65, 67), and table 1 indicates 48 examined (L65).

The authors write (L32) about "wintersurvivaltraining" but do not provide data on the time of year and geographical latitude where testing was performed – but it is known that the level of cortisol in the cold depends on the level of illumination. Also, in people with heavy physical activity (athletes of winter cyclical sports), cortisol depends on the season and on the duration of pre-exercise.

The test provides data on the intensity of the total energy consumption of the subjects using a method that is not the most popular for scientific literature. The technique used is primarily intended for monitoring energy consumption for personal use and its error is relatively high – especially in the extreme conditions in which the test subjects were. The number of parameters in the basic package of the program is quite scarce. Despite the reference (18) in the text, there are reasonable doubts about the accuracy of the method of estimating energy consumption by the method used by the authors at a high level and intensity of physical exertion.

 Due to the fact that consumption in the described experiment could be at the level of "basal metabolic rate", in this connection, there could be persons in the groups whose consumption was higher and lower than this indicator, and this could affect the results of the study.

The materials and methods section is not clearly described. From the description, it is still not clear how long the subjects were at negative temperature daily during different phases of the examination.

In addition, it is not clear from the description what specific temperature the subjects were in the air, and at what temperature the research testing was carried out indoors, as well as what time after entering the room the testing was carried out.

Table 3 requires clarification and elaboration for its understanding by readers. 

It is not entirely clear in general how a 20 m race can be considered as a method of assessing endurance – since 20 meters is the work of the creatine phosphate energy supply system – I would like to have more detailed explanations in the text on this score.

The study participants were instructed to make maximum effort as quickly as possible with the verbal support of the testing staff L134, but it turns out that verbally the participants of the experiment could motivate the subjects in the study when performing the test – this is unacceptable for this kind of research.

In the section conclusion of the work, in the opinion of the reviewer according to the text, there is a lack of description of the novelty of the study. It is obvious that with daily energy consumption, the huge amount of consumption proposed in the study is insufficient and will lead to a shortage of energy. And it is obvious that after a while this deficit will close. Thus, I would like the authors to reflect more clearly in the conclusion what is the novelty of the data obtained in the research, especially since it is presented more clearly in abstract.

Author Response

Reviewer1
Comments and Suggestions for Authors
The research is undoubtedly interesting, relevant and corresponds to the direction of the journal. At the same time, the reviewer has a number of comments and suggestions.

A: We appreciate that you found our study interesting and relevant. We agree that the novelty of the study needs to be emphasized and for that, we have modified the introduction (L48-52, L57-59).  

The text repeatedly states that 46 soldiers were examined (L65, 67), and table 1 indicates 48 examined (L65).

A: Thank you for your careful review. We have corrected the total number of soldiers to the text. As we used the mixed model, different number of subjects were analyzed in each test due to some missing data. Moreover, clarifying sentences have been added to methods (L184-186).  

The authors write (L32) about "winter survival training" but do not provide data on the time of year and geographical latitude where testing was performed – but it is known that the level of cortisol in the cold depends on the level of illumination. Also, in people with heavy physical activity (athletes of winter cyclical sports), cortisol depends on the season and on the duration of pre-exercise.

A: The study was conducted in Lapland north from Arctic Circle (location 67°24’54” N, 26°35’26” E) in early April, and this information has been added to the methods (L90-91). The studied period was 8 days, and the illumination increased day by day at that time of the year (sunrise 0553-0614 am, sunset 0822-0837 pm). Variation of cortisol levels according to temperature and illumination is an interesting point of view, and needs consideration in future. 

The test provides data on the intensity of the total energy consumption of the subjects using a method that is not the most popular for scientific literature. The technique used is primarily intended for monitoring energy consumption for personal use and its error is relatively high – especially in the extreme conditions in which the test subjects were. The number of parameters in the basic package of the program is quite scarce. Despite the reference (18) in the text, there are reasonable doubts about the accuracy of the method of estimating energy consumption by the method used by the authors at a high level and intensity of physical exertion.

A: We agree with your opinion. We are aware that there are always bias, random errors and, systematic errors, when measuring energy expenditure and intake. Heart rate variability is not the most accurate method, but considering the study protocol, golden standard method (double labeled water) was not possible. Furthermore, the cost of this isotope method limits its use. Validation studies by Smolander et al. (2011) and Robertson et al. (2016) found a moderate accuracy (7-10 %), and Montgomery et al. (2009) observed 13 % underestimation of energy expenditure by HRV system. These studies (Robertson et al. 2016, Montgomery et al. 2009) were carried out in free-living participants with moderate physical activity and no limitations of sleep and these conditions vary a lot from conditions of our study. However, measuring energy expenditure in an accurate way is very challenging in field conditions, but we have tried to get data from field and use our method being aware its limitations.

 Due to the fact that consumption in the described experiment could be at the level of "basal metabolic rate", in this connection, there could be persons in the groups whose consumption was higher and lower than this indicator, and this could affect the results of the study.

A: Thank you for your comment. We agree, that energy intake rates were low, and expenditure high especially in the field training phase. In most cases, energy intake was lower than their estimated basal metabolic rate, but overall the negative energy balance varies obviously between the participants. Intraindividual variation, depending on body mass, body composition, hormonal status etc. can affect the BMR and via that, energy expenditure levels. 

The materials and methods section is not clearly described. From the description, it is still not clear how long the subjects were at negative temperature daily during different phases of the examination.

A: Thank you for your notification. Weather was typical for late winter with changes in day and night temperatures. On sunny days, temperature raised rapidly above zero in the morning (sunrise 05.53-06.14), and declined below zero after sunset (20.22-2037). Daytime became longer day by day during spring
Unfortunately, we do not have exact hours, how long participants were at negative temperature, but we assume 12-16 hours per day. However, participants were adapted to cold temperature and winter conditions, and this has been added to the methods (L97-102). In Lapland, January and February are usually the coldest months of the year and in April, temperatures are more comfortable.

In addition, it is not clear from the description what specific temperature the subjects were in the air, and at what temperature the research testing was carried out indoors, as well as what time after entering the room the testing was carried out.

A: Thank you for your detailed revision. Since the study was part of a larger multidisciplinary research project, lot of different measurements were scheduled for the test day, starting from body composition and blood samples and finishing physical fitness and shooting tests. For the body composition measurements, participants arrived to the test room directly from the field, early in the morning (05 am), and the outdoor temperature was below zero. Fitness tests were carried out approximately at 11 am-01 pm, after few hours other (cognitive) testing in an indoor temperature between 19-21 degrees.

Table 3 requires clarification and elaboration for its understanding by readers.

A: We have corrected the table 3, by adding units and modifying the columns. Hopefully this clarifies the table.

It is not entirely clear in general how a 20 m race can be considered as a method of assessing endurance – since 20 meters is the work of the creatine phosphate energy supply system – I would like to have more detailed explanations in the text on this score.

A: Thank you for the comment. We have now written in the methods: “Endurance performance was measured by a 20-m shuttle run test [25]. Participants were advised to run 20 meters back and forth with accelerating pace as long as possible. The test was finished, when participants were not able to keep the given pace or voluntarily dropped out. Maximal oxygen uptake values were calculated from test results as previously described [26].”  Based on the test protocol (increasing intensity to maximum), we suggested that muscle glycogen was the main substrate for the performance, with increasing glycogen use towards the end of the test. 

 The study participants were instructed to make maximum effort as quickly as possible with the verbal support of the testing staff L134, but it turns out that verbally the participants of the experiment could motivate the subjects in the study when performing the test – this is unacceptable for this kind of research.

A: Thank you for your view. From our experience, verbal commands and encouragements have been done systematically in maximal force production test trials and documented by several researchers over decades (Häkkinen et al. (1985) 2001, Ojanen et al. (2020)). 
Häkkinen K, Pakarinen A, Kraemer WJ, Häkkinen A, Valkeinen H, Alen M. Selective muscle hypertrophy, changes in EMG and force, and serum hormones during strength training in older women. J Appl Physiol (1985). 2001 Aug;91(2):569-80. doi: 10.1152/jappl.2001.91.2.569. 
Ojanen T, Kyröläinen H, Kozharskaya E, Häkkinen K. Changes in strength and power performance and serum hormone concentrations during 12 weeks of task-specific or strength training in conscripts. Physiol Rep. 2020 May;8(9):e14422. doi: 10.14814/phy2.14422. PMID: 32378340; PMCID: PMC7202986.)

In the section conclusion of the work, in the opinion of the reviewer according to the text, there is a lack of description of the novelty of the study. It is obvious that with daily energy consumption, the huge amount of consumption proposed in the study is insufficient and will lead to a shortage of energy. And it is obvious that after a while this deficit will close. Thus, I would like the authors to reflect more clearly in the conclusion what is the novelty of the data obtained in the research, especially since it is presented more clearly in abstract.

A: We modified the conclusion according to your suggestions, by emphasizing the meaning of recovery period and its outcomes in military performance (L384-387).

Reviewer 2 Report

se the attached file

Author Response

A: Thank you for your revision. We agree, that the list of methods might be monotonous, since we had numerous variables in our study. For that reason, we decided to describe shortly the method and add references, in which the method is carefully documented. In addition, we modified the table 2 by inserting the method used. We have also corrected spelling mistakes (L198).

Round 2

Reviewer 2 Report

none